# Approximation Algorithms for $\ell_0$-Low Rank Approximation

**Karl Bringmann**[1]
kbringma@mpi-inf.mpg.de

**Pavel Kolev**[1]*
pkolev@mpi-inf.mpg.de

**David P. Woodruff**[2]
dwoodruf@cs.cmu.edu

[1] Max Planck Institute for Informatics, Saarland Informatics Campus, Saarbrücken, Germany
[2] Department of Computer Science, Carnegie Mellon University

## Abstract

We study the $\ell_0$-Low Rank Approximation Problem, where the goal is, given an $m \times n$ matrix $A$, to output a rank-$k$ matrix $A'$ for which $\|A' - A\|_0$ is minimized. Here, for a matrix $B$, $\|B\|_0$ denotes the number of its non-zero entries. This NP-hard variant of low rank approximation is natural for problems with no underlying metric, and its goal is to minimize the number of disagreeing data positions.

We provide approximation algorithms which significantly improve the running time and approximation factor of previous work. For $k > 1$, we show how to find, in $\text{poly}(mn)$ time for every $k$, a rank $O(k \log(n/k))$ matrix $A'$ for which $\|A' - A\|_0 \le O(k^2 \log(n/k))$ OPT. To the best of our knowledge, this is the first algorithm with provable guarantees for the $\ell_0$-Low Rank Approximation Problem for $k > 1$, even for bicriteria algorithms.

For the well-studied case when $k = 1$, we give a $(2+\epsilon)$-approximation in *sublinear time*, which is impossible for other variants of low rank approximation such as for the Frobenius norm. We strengthen this for the well-studied case of binary matrices to obtain a $(1 + O(\psi))$-approximation in sublinear time, where $\psi = \text{OPT}/\|A\|_0$. For small $\psi$, our approximation factor is $1 + o(1)$.

## 1   Introduction

Low rank approximation of an $m \times n$ matrix $A$ is an extremely well-studied problem, where the goal is to replace the matrix $A$ with a rank-$k$ matrix $A'$ which well-approximates $A$, in the sense that $\|A - A'\|$ is small under some measure $\|\cdot\|$. Since any rank-$k$ matrix $A'$ can be written as $U \cdot V$, where $U$ is $m \times k$ and $V$ is $k \times n$, this allows for a significant parameter reduction. Namely, instead of storing $A$, which has $mn$ entries, one can store $U$ and $V$, which have only $(m+n)k$ entries in total. Moreover, when computing $Ax$, one can first compute $Vx$ and then $U(Vx)$, which takes $(m + n)k$ instead of $mn$ time. We refer the reader to several surveys [19, 24, 40] for references to the many results on low rank approximation.

We focus on approximation algorithms for the low-rank approximation problem, i.e. we seek to output a rank-$k$ matrix $A'$ for which $\|A - A'\| \le \alpha\|A - A_k\|$, where $A_k = \text{argmin}_{\text{rank}(B)=k}\|A - B\|$ is the best rank-k approximation to $A$, and the approximation ratio $\alpha$ is as small as possible. One of the most widely studied error measures is the Frobenius norm $\|A\|_F = (\sum_{i=1}^{m} \sum_{j=1}^{n} A_{i,j}^2)^{1/2}$, for which the optimal rank-k approximation can be obtained via the singular value decomposition (SVD). Using randomization and approximation, one can compute an $\alpha = 1 + \epsilon$-approximation, for any $\epsilon > 0$, in time much faster than the $\min(mn^2, mn^2)$ time required for computing the SVD, namely, in $O(\|A\|_0 + n \cdot \text{poly}(k/\epsilon))$ time [9, 26, 29], where $\|A\|_0$ denotes the number of non-zero entries

of $A$. For the Frobenius norm $\|A\|_0$ time is also a lower bound, as any algorithm that does not read nearly all entries of $A$ might not read a very large entry, and therefore cannot achieve a relative error approximation.

The rank-$k$ matrix $A_k$ obtained by computing the SVD is also optimal with respect to any rotationally invariant norm, such as the operator and Schatten-$p$ norms. Thus, such norms can also be solved exactly in polynomial time. Recently, however, there has been considerable interest [10, 3, 32] in obtaining low rank approximations for NP-hard error measures such as the *entrywise* $\ell_p$-norm $\|A\|_p = \left( \sum_{i,j} |A_{i,j}|^p \right)^{1/p}$, where $p \geq 1$ is a real number. Note that for $p < 1$, this is not a norm, though it is still a well-defined quantity. For $p = \infty$, this corresponds to the max-norm or Chebyshev norm. It is known that one can achieve a $\mathrm{poly}(k \log(mn))$-approximation in $\mathrm{poly}(mn)$ time for the low-rank approximation problem with entrywise $\ell_p$-norm for every $p \geq 1$ [36, 8].

## 1.1 $\ell_0$-Low Rank Approximation

A natural variant of low rank approximation which the results above do not cover is that of $\ell_0$-*low rank approximation*, where the measure $\|A\|_0$ is the number of non-zero entries. In other words, we seek a rank-$k$ matrix $A'$ for which the number of entries $(i,j)$ with $A'_{i,j} \neq A_{i,j}$ is as small as possible. Letting $\mathrm{OPT} = \min_{\mathrm{rank}(B)=k} \sum_{i,j} \delta(A_{i,j} \neq A'_{i,j})$, where $\delta(A_{i,j} \neq A'_{i,j}) = 1$ if $A_{i,j} \neq A'_{i,j}$ and 0 otherwise, we would like to output a rank-$k$ matrix $A'$ for which there are at most $\alpha\,\mathrm{OPT}$ entries $(i,j)$ with $A'_{i,j} \neq A_{i,j}$. Approximation algorithms for this problem are essential since solving the problem exactly is NP-hard [12, 14], even when $k = 1$ and $A$ is a binary matrix.

The $\ell_0$-low rank approximation problem is quite natural for problems with no underlying metric, and its goal is to minimize the number of disagreeing data positions with a low rank matrix. Indeed, this error measure directly answers the following question: if we are allowed to ignore some data - outliers or anomalies - what is the best low-rank model we can get? One well-studied case is when $A$ is binary, but $A'$ and its factors $U$ and $V$ need not necessarily be binary. This is called unconstrained Binary Matrix Factorization in [18], which has applications to association rule mining [20], biclustering structure identification [42, 43], pattern discovery for gene expression [34], digits reconstruction [25], mining high-dimensional discrete-attribute data [21, 22], market based clustering [23], and document clustering [43]. There is also a body of work on Boolean Matrix Factorization which restricts the factors to also be binary, which is referred to as constrained Binary Matrix Factorization in [18]. This is motivated in applications such as classifying text documents and there is a large body of work on this, see, e.g. [28, 31].

The $\ell_0$-low rank approximation problem coincides with a number of problems in different areas. It exactly coincides with the famous matrix rigidity problem over the reals, which asks for the minimal number OPT of entries of $A$ that need to be changed in order to obtain a matrix of rank at most $k$. The matrix rigidity problem is well-studied in complexity theory [15, 16, 39] and parameterized complexity [13]. These works are not directly relevant here as they do not provide approximation algorithms. There are also other variants of $\ell_0$-low rank approximation, corresponding to cases such as when $A$ is binary, $A' = UV$ is required to have binary factors $U$ and $V$, and multiplication is either performed over a binary field [41, 17, 12, 30], or corresponds to an OR of ANDs. The latter is known as the Boolean model [4, 12, 27, 33, 35, 38]. These different notions of inner products lead to very different algorithms and results for the $\ell_0$-low rank approximation problem. However, all these models coincide in the special and important case in which $A$ is binary and $k = 1$. This case was studied in [20, 34, 18], as their algorithm for $k = 1$ forms the basis for their successful heuristic for general $k$, e.g. the PROXIMUS technique [20].

Another related problem is robust PCA [6], in which there is an underlying matrix $A$ that can be written as a low rank matrix $L$ plus a sparse matrix $S$ [7]. Candès et al. [7] argue that both components are of arbitrary magnitude, and we do not know the locations of the non-zeros in $S$ nor how many there are. Moreover, grossly corrupted observations are common in image processing, web data analysis, and bioinformatics where some measurements are *arbitrarily* corrupted due to occlusions, malicious tampering, or sensor failures. Specific scenarios include video surveillance, face recognition, latent semantic indexing, and ranking of movies, books, etc. [7]. These problems have the common theme of being an arbitrary magnitude sparse perturbation to a low rank matrix with no natural underlying metric, and so the $\ell_0$-error measure (which is just the Hamming distance, or number of disagreements) is appropriate. In order to solve robust PCA in practice, Candès et al. [7]

relaxed the $\ell_0$-error measure to the $\ell_1$-norm. Understanding theoretical guarantees for solving the original $\ell_0$-problem is of fundamental importance, and we study this problem in this paper.

Finally, interpreting $0^0$ as 0, the $\ell_0$-low rank approximation problem coincides with the aforementioned notion of entrywise $\ell_p$-approximation when $p = 0$. It is not hard to see that previous work [8] for general $p \geq 1$ fails to give any approximation factor for $p = 0$. Indeed, critical to their analysis is the scale-invariance property of a norm, which does not hold for $p = 0$ since $\ell_0$ is not a norm.

## 1.2  Our Results

We provide approximation algorithms for the $\ell_0$-low rank approximation problem which significantly improve the running time or approximation factor of previous work. In some cases our algorithms even run in *sublinear time*, i.e., faster than reading all non-zero entries of the matrix. This is provably impossible for other measures such as the Frobenius norm and more generally, any $\ell_p$-norm for $p > 0$. For $k > 1$, our approximation algorithms are, to the best of our knowledge, the first with provable guarantees for this problem.

First, for $k = 1$, we significantly improve the polynomial running time of previous $(2 + \epsilon)$-approximations for this problem. The best previous algorithm due to Jiang et al. [18] was based on the observation that there exists a column $u$ of $A$ spanning a 2-approximation. Therefore, solving the problem $\min_v \|A - uv\|_0$ for each column $u$ of $A$ yields a 2-approximation, where for a matrix $B$ the measure $\|B\|_0$ counts the number of non-zero entries. The problem $\min_v \|A - uv\|_0$ decomposes into $\sum_i \min_i \|A_{:,i} - v_i u\|_0$, where $A_{:,i}$ is the $i$-th column of $A$, and $v_i$ the $i$-th entry of vector $v$. The optimal $v_i$ is the mode of the ratios $A_{i,j}/u_j$, where $j$ ranges over indices in $\{1, 2, \ldots, m\}$ with $u_j \neq 0$. As a result, one can find a rank-1 matrix $uv^T$ providing a 2-approximation in $O(\|A\|_0 n)$ time, which was the best known running time. Somewhat surprisingly, we show that one can achieve *sublinear time* for solving this problem. Namely, we obtain a $(2 + \epsilon)$-approximation in $(m + n)\operatorname{poly}(\epsilon^{-1}\psi^{-1}\log(mn))$ time, for any $\epsilon > 0$, where $\psi = \operatorname{OPT}/\|A\|_0$. This significantly improves upon the earlier $O(\|A\|_0 n)$ time for not too small $\epsilon$ and $\psi$. Our result should be contrasted to Frobenius norm low rank approximation, for which $\Omega(\|A\|_0)$ time is required even for $k = 1$, as otherwise one might miss a very large entry in $A$. Since $\ell_0$-low rank approximation is insensitive to the magnitude of entries of $A$, we bypass this general impossibility result.

Next, still considering the case of $k = 1$, we show that if the matrix $A$ is binary, a well-studied case coinciding with the abovementioned $GF(2)$ and Boolean models, we obtain an approximation algorithm parameterized in terms of the ratio $\psi = \operatorname{OPT}/\|A\|_0$, showing it is possible in time $(m + n)\psi^{-1}\operatorname{poly}(\log(mn))$ to obtain a $(1 + O(\psi))$-approximation. Note that our algorithm is again sublinear, unlike all algorithms in previous work. Moreover, when $A$ is itself very well approximated by a low rank matrix, then $\psi$ may actually be sub-constant, and we obtain a significantly better $(1 + o(1))$-approximation than the previous best known 2-approximations. Thus, we simultaneously improve the running time and approximation factor. We also show that the running time of our algorithm is optimal up to $\operatorname{poly}(\log(mn))$ factors by proving that any $(1 + O(\psi))$-approximation succeeding with constant probability must read $\Omega((m + n)\psi^{-1})$ entries of $A$ in the worst case.

Finally, for arbitrary $k > 1$, we first give an impractical algorithm that runs in time $n^{O(k)}$ and achieves an $\alpha = \operatorname{poly}(k)$-approximation. To the best of our knowledge this is the first approximation algorithm for the $\ell_0$-low rank approximation problem with any non-trivial approximation factor. To make our algorithm practical, we reduce the running time to $\operatorname{poly}(mn)$, with an exponent independent of $k$, if we allow for a bicriteria solution. In particular, we allow the algorithm to output a matrix $A'$ of somewhat larger rank $O(k \log(n/k))$, for which $\|A - A'\|_0 \leq O(k^2 \log(n/k)) \min_{\operatorname{rank}(B)=k} \|A - B\|_0$. Although we do not obtain rank exactly $k$, many of the motivations for finding a low rank approximation, such as reducing the number of parameters and fast matrix-vector product, still hold if the output rank is $O(k \log(n/k))$. We are not aware of any alternative algorithms which achieve $\operatorname{poly}(mn)$ time and any provable approximation factor, even for bicriteria solutions.

## 2  Preliminaries

For an matrix $A \in \mathbb{A}^{m \times n}$ with entries $A_{i,j}$, we write $A_{i,:}$ for its $i$-th row and $A_{:,j}$ for its $j$-th column.

**Input Formats** We always assume that we have random access to the entries of the given matrix $A$, i.e. we can read any entry $A_{i,j}$ in constant time. For our sublinear time algorithms we need more efficient access to the matrix, specifically the following two variants:

(1) We say that we are given $A$ *with column adjacency arrays* if we are given arrays $B_1, \ldots, B_n$ and lengths $\ell_1, \ldots, \ell_n$ such that for any $1 \leq k \leq \ell_j$ the pair $B_j[k] = (i, A_{i,j})$ stores the row $i$ containing the $k$-th nonzero entry in column $j$ as well as that entry $A_{i,j}$. This is a standard representation of matrices used in many applications. Note that given only these adjacency arrays $B_1, \ldots, B_n$, in order to access any entry $A_{i,j}$ we can perform a binary search over $B_j$, and hence random access to any matrix entry is in time $O(\log n)$. Moreover, we assume to have random access to matrix entries in constant time, and note that this is optimistic by at most a factor $O(\log n)$.

(2) We say that we are given matrix $A$ *with row and column sums* if we can access the numbers $\sum_j A_{i,j}$ for $i \in [m]$ and $\sum_i A_{i,j}$ for $j \in [n]$ in constant time (and, as always, access any entry $A_{i,j}$ in constant time). Notice that storing the row and column sums takes $O(n+m)$ space, and thus while this might not be standard information it is very cheap to store.

We show that the first access type even allows to sample from the set of nonzero entries uniformly in constant time.

**Lemma 1.** *Given a matrix $A \in \mathbb{R}^{m \times n}$ with column adjacency arrays, after $O(n)$ time preprocessing we can sample a uniformly random nonzero entry $(i, j)$ from $A$ in time $O(1)$.*

The proof of this lemma, as well as most other proofs in this extended abstract, can be found in the full version of the paper.

## 3 Algorithms for Real $\ell_0$-rank-$k$

Given a matrix $A \in \mathbb{R}^{m \times n}$, the $\ell_0$-rank-$k$ problem asks to find a matrix $A'$ with rank $k$ such that the difference between $A$ and $A'$ measured in $\ell_0$ norm is minimized. We denote the optimum value by

$$\mathrm{OPT}^{(k)} \stackrel{\text{def}}{=} \min_{\mathrm{rank}(A')=k} \|A - A'\|_0 = \min_{U \in \mathbb{R}^{m \times k}, V \in \mathbb{R}^{k \times n}} \|A - UV\|_0. \tag{1}$$

In this section, we establish several new results on the $\ell_0$-rank-$k$ problem. In Subsection 3.1, we prove a structural lemma that shows the existence of $k$ columns which provide a $(k+1)$-approximation to $\mathrm{OPT}^{(k)}$, and we also give an $\Omega(k)$-approximation lower bound for any algorithm that selects $k$ columns from the input matrix $A$. In Subsection 3.2, we give an approximation algorithm that runs in $\mathrm{poly}(n^k, m)$ time and achieves an $O(k^2)$-approximation. To the best of our knowledge, this is the first algorithm with provable non-trivial approximation guarantees. In Subsection 3.3, we design a practical algorithm that runs in $\mathrm{poly}(n, m)$ time with an exponent independent of $k$, if we allow for a bicriteria solution.

### 3.1 Structural Results

We give a new structural result for $\ell_0$ showing that any matrix $A$ contains $k$ columns which provide a $(k+1)$-approximation for the $\ell_0$-rank-$k$ problem (1).

**Lemma 2.** *Let $A \in \mathbb{R}^{m \times n}$ be a matrix and $k \in [n]$. There is a subset $J^{(k)} \subset [n]$ of size $k$ and a matrix $Z \in \mathbb{R}^{k \times n}$ such that $\|A - A_{:,J^{(k)}} Z\|_0 \leq (k+1)\mathrm{OPT}^{(k)}$.*

*Proof.* Let $Q^{(0)}$ be the set of columns $j$ with $UV_{:,j} = 0$, and let $R^{(0)} \stackrel{\text{def}}{=} [n] \setminus Q^{(0)}$. Let $S^{(0)} \stackrel{\text{def}}{=} [n]$, $T^{(0)} \stackrel{\text{def}}{=} \emptyset$. We split the value $\mathrm{OPT}^{(k)}$ into $\mathrm{OPT}(S^{(0)}, R^{(0)}) \stackrel{\text{def}}{=} \|A_{S^{(0)},R^{(0)}} - UV_{S^{(0)},R^{(0)}}\|_0$ and $\mathrm{OPT}(S^{(0)}, Q^{(0)}) \stackrel{\text{def}}{=} \|A_{S^{(0)},Q^{(0)}} - UV_{S^{(0)},Q^{(0)}}\|_0 = \|A_{S^{(0)},Q^{(0)}}\|_0$.

Suppose $\mathrm{OPT}(S^{(0)}, R^{(0)}) \geq |S^{(0)}||R^{(0)}|/(k+1)$. Then, for any subset $J^{(k)}$ it follows that $\min_Z \|A - A_{S^{(0)},J^{(k)}} Z\|_0 \leq |S^{(0)}||R^{(0)}| + \|A_{S^{(0)},Q^{(0)}}\|_0 \leq (k+1)\mathrm{OPT}^{(k)}$. Otherwise, there is a column $i^{(1)}$ such that $\|A_{S^{(0)},i^{(1)}} - (UV)_{S^{(0)},i^{(1)}}\|_0 \leq \mathrm{OPT}(S^{(0)}, R^{(0)})/|R^{(0)}| \leq \mathrm{OPT}^{(k)}/|R^{(0)}|$.

Let $T^{(1)}$ be the set of indices on which $(UV)_{S^{(0)}, i^{(1)}}$ and $A_{S^{(0)}, i^{(1)}}$ disagree, and similarly $S^{(1)} \overset{\text{def}}{=} S^{(0)} \backslash T^{(1)}$ on which they agree. Then we have $|T^{(1)}| \leq \text{OPT}^{(k)}/|R^{(0)}|$. Hence, in the submatrix $T^{(1)} \times R^{(0)}$ the total error is at most $|T^{(1)}| \cdot |R^{(0)}| \leq \text{OPT}^{(k)}$. Let $R^{(1)}, D^{(1)}$ be a partitioning of $R^{(0)}$ such that $A_{S^{(1)}, j}$ is linearly dependent on $A_{S^{(1)}, i^{(1)}}$ iff $j \in D^{(1)}$. Then by selecting column $A_{:, i^{(1)}}$ the incurred cost on matrix $S^{(1)} \times D^{(1)}$ is zero. For the remaining submatrix $S^{(\ell)} \times R^{(\ell)}$, we perform a recursive call of the algorithm.

We make at most $k$ recursive calls, on instances $S^{(\ell)} \times R^{(\ell)}$ for $\ell \in \{0, \ldots, k-1\}$. In the $\ell^{th}$ iteration, either $\text{OPT}(S^{(\ell)}, R^{(\ell)}) \geq |S^{(\ell)}||R^{(\ell)}|/(k+1-\ell)$ and we are done, or there is a column $i^{(\ell+1)}$ which partitions $S^{(\ell)}$ into $S^{(\ell+1)}, T^{(\ell+1)}$ and $R^{(\ell)}$ into $R^{(\ell+1)}, D^{(\ell+1)}$ such that $|S^{(\ell+1)}| \geq m \cdot \prod_{i=0}^{\ell} (1 - \frac{1}{k+1-i}) = \frac{k-\ell}{k+1} \cdot m$ and for every $j \in D^{(\ell)}$ the column $A_{S^{(\ell+1)}, j}$ belongs to the span of $\{A_{S^{(\ell+1)}, i^{(t)}}\}_{t=1}^{\ell+1}$.

Suppose we performed $k$ recursive calls. We show now that the incurred cost in submatrix $S^{(k)} \times R^{(k)}$ is at most $\text{OPT}(S^{(k)}, R^{(k)}) \leq \text{OPT}^{(k)}$. By construction, the sub-columns $\{A_{S^{(k)}, i}\}_{i \in I^{(k)}}$ are linearly independent, where $I^{(k)} = \{i^{(1)}, \ldots, i^{(k)}\}$ is the set of the selected columns, and $A_{S^{(k)}, I^{(k)}} = (UV)_{S^{(k)}, I^{(k)}}$. Since $\text{rank}(A_{S^{(k)}, I^{(k)}}) = k$, it follows that $\text{rank}(U_{S^{(k)}, :}) = k$, $\text{rank}(V_{:, I^{(k)}}) = k$ and the matrix $V_{:, I^{(k)}} \in \mathbb{R}^{k \times k}$ is invertible. Hence, for matrix $Z = (V_{:, I^{(k)}})^{-1} V_{:, R^k}$ we have $\text{OPT}(S^{(k)}, R^{(k)}) = \|A_{S^k, R^k} - A_{S^k, I^k} Z\|_0$.

The statement follows by noting that the recursive calls accumulate a total cost of at most $k \cdot \text{OPT}^{(k)}$ in the submatrices $T^{(\ell+1)} \times R^{(\ell)}$ for $\ell \in \{0, 1, \ldots, k-1\}$, as well as cost at most $\text{OPT}^{(k)}$ in submatrix $S^{(k)} \times R^{(k)}$. $\square$

We also show that any algorithm that selects $k$ columns of a matrix $A$ incurs at least an $\Omega(k)$-approximation for the $\ell_0$-rank-$k$ problem.

**Lemma 3.** *Let $k \leq n/2$. Suppose $A = (G_{k \times n}; I_{n \times n}) \in \mathbb{R}^{(n+k) \times n}$ is a matrix composed of a Gaussian random matrix $G \in \mathbb{R}^{k \times n}$ with $G_{i,j} \sim N(0, 1)$ and identity matrix $I_{n \times n}$. Then for any subset $J^{(k)} \subset [n]$ of size $k$, we have $\min_{Z \in \mathbb{R}^{k \times n}} \|A - A_{:, J^{(k)}} Z\|_0 = \Omega(k) \cdot \text{OPT}^{(k)}$.*

### 3.2 Basic Algorithm

We give an impractical algorithm that runs in $\text{poly}(n^k, m)$ time and achieves an $O(k^2)$-approximation. To the best of our knowledge this is the first approximation algorithm for the $\ell_0$-rank-$k$ problem with non-trivial approximation guarantees.

**Theorem 4.** *Given $A \in \mathbb{R}^{m \times n}$ and $k \in [n]$ we can compute in $O(n^{k+1} m^2 k^{\omega+1})$ time a set of $k$ indices $J^{(k)} \subset [n]$ and a matrix $Z \in \mathbb{R}^{k \times n}$ such that $\|A - A_{:, J^{(k)}} Z\|_0 \leq O(k^2) \cdot \text{OPT}^{(k)}$.*

Our result relies on a subroutine by Berman and Karpinski [5] (attributed also to Kannan in that paper) which given a matrix $U$ and a vector $b$ approximates $\min_x \|Ux - b\|_0$ in polynomial time. Specifically, we invoke in our algorithm the following variant of this result established by Alon, Panigrahy, and Yekhanin [2].

**Theorem 5.** *[2] There is an algorithm that given $A \in \mathbb{R}^{m \times k}$ and $b \in \mathbb{R}^m$ outputs in $O(m^2 k^{\omega+1})$ time a vector $z \in \mathbb{R}^k$ such that w.h.p. $\|Az - b\|_0 \leq k \cdot \min_x \|Ax - b\|_0$.*

### 3.3 Bicriteria Algorithm

Our main contribution in this section is to design a practical algorithm that runs in $\text{poly}(n, m)$ time with an exponent independent of $k$, if we allow for a bicriteria solution.

**Theorem 6.** *Given $A \in \mathbb{R}^{m \times n}$ and $k \in [1, n]$, there is an algorithm that in expected $\text{poly}(m, n)$ time outputs a subset of indices $J \subset [n]$ with $|J| = O(k \log(n/k))$ and a matrix $Z \in \mathbb{R}^{|J| \times n}$ such that $\|A - A_{:, J} Z\|_0 \leq O(k^2 \log(n/k)) \cdot \text{OPT}^{(k)}$.*

The structure of the proof follows a recent approximation algorithm [8, Algorithm 3] for the $\ell_p$-low rank approximation problem, for any $p \geq 1$. We note that the analysis of [8, Theorem 7] is missing an

$O(\log^{1/p} n)$ approximation factor, and naïvely provides an $O(k \log^{1/p} n)$-approximation rather than the stated $O(k)$-approximation. Further, it might be possible to obtain an efficient algorithm yielding an $O(k^2 \log k)$-approximation for Theorem 6 using unpublished techniques in [37]; we leave the study of obtaining the optimal approximation factor to future work.

There are two critical differences with the proof of [8, Theorem 7]. We cannot use the earlier [8, Theorem 3] which shows that any matrix $A$ contains $k$ columns which provide an $O(k)$-approximation for the $\ell_p$-low rank approximation problem, since that proof requires $p \geq 1$ and critically uses scale-invariance, which does not hold for $p = 0$. Our combinatorial argument in Lemma 2 seems fundamentally different than the maximum volume submatrix argument in [8] for $p \geq 1$.

Second, unlike for $\ell_p$-regression for $p \geq 1$, the $\ell_0$-regression problem $\min_x \|Ux - b\|_0$ given a matrix $U$ and vector $b$ is not efficiently solvable since it corresponds to a nearest codeword problem, which is NP-hard [1]. Thus, we resort to an approximation algorithm for $\ell_0$-regression, based on ideas for solving the nearest codeword problem in [2, 5].

Note that $\mathrm{OPT}^{(k)} \leq \|A\|_0$. Since there are only $mn + 1$ possibilities of $\mathrm{OPT}^{(k)}$, we can assume we know $\mathrm{OPT}^{(k)}$ and we can run the Algorithm 1 below for each such possibility, obtaining a rank-$O(k \log n)$ solution, and then outputting the solution found with the smallest cost. This can be further optimized by forming instead $O(\log(mn))$ guesses of $\mathrm{OPT}^{(k)}$. One of these guesses is within a factor of 2 from the true value of $\mathrm{OPT}^{(k)}$, and we note that the following argument only needs to know $\mathrm{OPT}^{(k)}$ up to a factor of 2.

We start by defining the notion of approximate coverage, which is different than the corresponding notion in [8] for $p \geq 1$, due to the fact that $\ell_0$-regression cannot be efficiently solved. Consequently, approximate coverage for $p = 0$ cannot be efficiently tested. Let $Q \subseteq [n]$ and $M = A_{:,Q}$ be an $m \times |Q|$ submatrix of $A$. We say that a column $M_{:,i}$ is $(S, Q)$-*approximately covered* by a submatrix $M_{:,S}$ of $M$, if $|S| = 2k$ and $\min_x \|M_{:,S}x - M_{:,i}\|_0 \leq \frac{100(k+1)\mathrm{OPT}^{(k)}}{|Q|}$.

**Lemma 7.** *(Similar to [8, Lemma 6], but using Lemma 2) Let $Q \subseteq [n]$ and $M = A_{:,Q}$ be a submatrix of $A$. Suppose we select a subset $R$ of $2k$ uniformly random columns of $M$. Then with probability at least $1/3$, at least a $1/10$ fraction of the columns of $M$ are $(R, Q)$-approximately covered.*

*Proof.* To show this, as in [8], consider a uniformly random column index $i$ not in the set $R$. Let $T \overset{\mathrm{def}}{=} R \cup \{i\}$ and $\eta \overset{\mathrm{def}}{=} \min_{\mathrm{rank}(B)=k} \|M_{:,T} - B\|_0$. Since $T$ is a uniformly random subset of $2k + 1$ columns of $M$, $\mathbb{E}_T \eta \leq \frac{(2k+1)\mathrm{OPT}_M^{(k)}}{|Q|} \leq \frac{(2k+1)\mathrm{OPT}^{(k)}}{|Q|}$. Let $\mathcal{E}_1$ be the event $\eta \leq \frac{10(2k+1)\mathrm{OPT}^{(k)}}{|Q|}$. Then, by a Markov bound, $\Pr[\mathcal{E}_1] \geq 9/10$.

Fix a configuration $T = R \cup \{i\}$ and let $L(T) \subset T$ be the subset guaranteed by Lemma 2 such that $|L(T)| = k$ and $\min_X \|M_{:,L(T)}X - M_{:,T}\|_0 \leq (k+1) \min_{\mathrm{rank}(B)=k} \|M_{:,T} - B\|_0$. Notice that $\mathbb{E}_i \left[ \min_x \|M_{:,L(T)}x - M_{:,i}\|_0 \mid T \right] = \frac{1}{2k+1} \min_X \|M_{:,L(T)}X - M_{:,T}\|_0$, and thus by the law of total probability we have $\mathbb{E}_T \left[ \min_x \|M_{:,L(T)}x - M_{:,i}\|_0 \right] \leq \frac{(k+1)\eta}{2k+1}$.

Let $\mathcal{E}_2$ denote the event that $\min_x \|M_{:,L}x - M_{:,i}\|_0 \leq \frac{10(k+1)\eta}{2k+1}$. By a Markov bound, $\Pr[\mathcal{E}_2] \geq 9/10$.

Further, as in [8], let $\mathcal{E}_3$ be the event that $i \notin L$. Observe that there are $\binom{k+1}{k}$ ways to choose a subset $R' \subset T$ such that $|R'| = 2k$ and $L \subset R'$. Since there are $\binom{2k+1}{2k}$ ways to choose $R'$, it follows that $\Pr[L \subset R \mid T] = \frac{k+1}{2k+1} > 1/2$. Hence, by the law of total probability, we have $\Pr[\mathcal{E}_3] > 1/2$.

As in [8], $\Pr[\mathcal{E}_1 \wedge \mathcal{E}_2 \wedge \mathcal{E}_3] > 2/5$, and conditioned on $\mathcal{E}_1 \wedge \mathcal{E}_2 \wedge \mathcal{E}_3$, $\min_x \|M_{:,R}x - M_{:,i}\|_0 \leq \min_x \|M_{:,L}x - M_{:,i}\|_0 \leq \frac{10(k+1)\eta}{2k+1} \leq \frac{100(k+1)\mathrm{OPT}^{(k)}}{|Q|}$, where the first inequality uses that $L$ is a subset of $R$ given $\mathcal{E}_3$, and so the regression cost cannot decrease, while the second inequality uses the occurrence of $\mathcal{E}_2$ and the final inequality uses the occurrence of $\mathcal{E}_1$.

As in [8], if $Z_i$ is an indicator random variable indicating whether $i$ is approximately covered by $R$, and $Z = \sum_{i \in Q} Z_i$, then $\mathbb{E}_R[Z] \geq \frac{2|Q|}{5}$ and $\mathbb{E}_R[|Q| - Z] \leq \frac{3|Q|}{5}$. By a Markov bound, $\Pr[|Q| - Z \geq \frac{9|Q|}{10}] \leq \frac{2}{3}$. Thus, probability at least $1/3$, at least a $1/10$ fraction of the columns of $M$ are $(R, Q)$-approximately covered. $\square$

---

**Algorithm 1** Selecting $O(k \log(n/k))$ columns of $A$.

---

**Require:** An integer $k$, and a matrix $A$.
**Ensure:** $O(k \log(n/k))$ columns of $A$
APPROXIMATELYSELECTCOLUMNS $(k, A)$:
   **if** number of columns of $A \leq 2k$ **then**
      **return** all the columns of $A$
   **else**
      **repeat**
         Let $R$ be a set of $2k$ uniformly random columns of $A$
      **until** at least $(1/10)$-fraction columns of $A$ are nearly approximately covered
      Let $A_{\overline{R}}$ be the columns of $A$ not nearly approximately covered by $R$
      **return** $R \cup$ APPROXIMATELYSELECTCOLUMNS$(k, A_{\overline{R}})$
   **end if**

---

Given Lemma 7, we are ready to prove Theorem 6. As noted above, a key difference with the corresponding [8, Algorithm 3] for $\ell_p$ and $p \geq 1$, is that we cannot efficiently test if a column $i$ is approximately covered by a set $R$. We will instead again make use of Theorem 5.

*Proof of Theorem 6.* The computation of matrix $Z$ force us to relax the notion of $(R, Q)$-approximately covered to the notion of $(R, Q)$-*nearly-approximately covered* as follows: we say that a column $M_{:,i}$ is $(R, Q)$-*nearly-approximately covered* if, the algorithm in Theorem 5 returns a vector $z$ such that $\|M_{:,R}z - M_{:,i}\|_0 \leq \frac{100(k+1)^2 \text{OPT}^{(k)}}{|Q|}$. By the guarantee of Theorem 5, if $M_{:,i}$ is $(R, Q)$-approximately covered then it is also w.h.p. $(R, Q)$-nearly-approximately covered.

Suppose Algorithm 1 makes $t$ iterations and let $A_{:,\cup_{i=1}^t R_i}$ and $Z$ be the resulting solution. We bound now its cost. Let $B_0 = [n]$, and consider the $i$-th iteration of Algorithm 1. We denote by $R_i$ a set of $2k$ uniformly random columns of $B_{i-1}$, by $G_i$ a set of columns that is $(R_i, B_{i-1})$-nearly-approximately covered, and by $B_i = B_{i-1} \backslash \{G_i \cup R_i\}$ a set of the remaining columns. By construction, $|G_i| \geq |B_{i-1}|/10$ and $|B_i| \leq \frac{9}{10}|B_{i-1}| - 2k < \frac{9}{10}|B_{i-1}|$. Since Algorithm 1 terminates when $B_{t+1} \leq 2k$, we have $2k < |B_t| < (1 - \frac{1}{10})^t n$, and thus the number of iterations $t \leq 10 \log(n/2k)$. By construction, $|G_i| = (1 - \alpha_i)|B_{i-1}|$ for some $\alpha_i \leq 9/10$, and so $\sum_{i=1}^t \frac{|G_i|}{|B_{i-1}|} \leq t \leq 10 \log \frac{n}{2k}$. Since $\min_{x^{(j)}} \|A_{:,R_i}x^{(j)} - A_{:,j}\|_0 \leq \frac{100(k+1)^2 \text{OPT}^{(k)}}{|B_{i-1}|}$, we have $\sum_{i=1}^t \sum_{j \in G_i} \|A_{:,R_i}z^{(j)} - A_{:,j}\|_0 \leq \sum_{i=1}^t \sum_{j \in G_i} k \cdot \min_{x^{(j)}} \|A_{:,R_i}x^{(j)} - A_{:,j}\|_0 \leq O\left(k^2 \cdot \log \frac{n}{2k}\right) \cdot \text{OPT}^{(k)}$.

By Lemma 7, the expected number of iterations of selecting a set $R_i$ such that $|G_i| \geq 1/10|B_{i-1}|$ is $O(1)$. Since the number of recursive calls $t$ is bounded by $O(\log(n/k))$, it follows by a Markov bound that Algorithm 1 chooses $O(k \log(n/k))$ columns in total. Since the approximation algorithm of Theorem 5 runs in polynomial time, our entire algorithm has expected polynomial time. $\qquad\square$

## 4 Algorithm for Real $\ell_0$-rank-1

Given a matrix $A \in \mathbb{R}^{m \times n}$, the $\ell_0$-rank-1 problem asks to find a matrix $A'$ with rank 1 such that the difference between $A$ and $A'$ measured in $\ell_0$ norm is minimized. We denote the optimum value by

$$\text{OPT}^{(1)} \stackrel{\text{def}}{=} \min_{\text{rank}(A')=1} \|A - A'\|_0 = \min_{u \in \mathbb{R}^m, \, v \in \mathbb{R}^n} \|A - uv^T\|_0. \qquad (2)$$

In the trivial case when $\text{OPT}^{(1)} = 0$, there is an optimal algorithm that runs in time $O(\|A\|_0)$ and finds the exact rank-1 decomposition $uv^T$ of a matrix $A$. In this work, we focus on the case when $\text{OPT}^{(1)} \geq 1$. We show that Algorithm 2 yields a $(2 + \epsilon)$-approximation factor and runs in nearly linear time in $\|A\|_0$, for any constant $\epsilon > 0$. Furthermore, a variant of our algorithm even runs in sublinear time, if $\|A\|_0$ is large and $\psi \stackrel{\text{def}}{=} \text{OPT}^{(1)}/\|A\|_0$ is not too small. In particular, we obtain time $o(\|A\|_0)$ when $\text{OPT}^{(1)} \geq (\epsilon^{-1} \log(mn))^4$ and $\|A\|_0 \geq n(\epsilon^{-1} \log(mn))^4$.

**Algorithm 2**

---

**Input:** $A \in \mathbb{R}^{m \times n}$ and $\epsilon \in (0, 0.1)$.

1. Partition the columns of $A$ into weight-classes $\mathcal{S} = \{S^{(0)}, \ldots, S^{(\log n + 1)}\}$ such that $S^{(0)}$ contains all columns $j$ with $\|A_{:,j}\|_0 = 0$ and $S^{(i)}$ contains all columns $j$ with $2^{i-1} \leq \|A_{:,j}\|_0 < 2^i$.

2. For each weight-class $S^{(i)}$ do:

    2.1 Sample a set $C^{(i)}$ of $\Theta(\epsilon^{-2} \log n)$ elements uniformly at random from $S^{(i)}$.

    2.2 Find a vector $z^{(j)} \in \mathbb{R}^n$ such that $\|A - A_{:,j}[z^{(j)}]^T\|_0 \leq \left(1 + \frac{\epsilon}{15}\right) \min_v \|A - A_{:,j} v^T\|_0$, for each column $A_{:,j} \in C^{(i)}$.

3. Compute a $(1 + \frac{\epsilon}{15})$-approximation $Y_j$ of $\|A - A_{:,j}[z^{(j)}]^T\|_0$ for every $j \in \bigcup_{i \in [|\mathcal{S}|]} C^{(i)}$.

**Return:** the pair $(A_{:,j}, z^{(j)})$ corresponding to the minimal value $Y_j$.

---

**Theorem 8.** *There is an algorithm that, given $A \in \mathbb{R}^{m \times n}$ with column adjacency arrays and* $\mathrm{OPT}^{(1)} \geq 1$*, and given $\epsilon \in (0, 0.1]$, runs w.h.p. in time $O\left(\left(\frac{n \log m}{\epsilon^2} + \min\left\{\|A\|_0,\ n + \psi^{-1} \frac{\log n}{\epsilon^2}\right\}\right) \frac{\log^2 n}{\epsilon^2}\right)$ and outputs a column $A_{:,j}$ and a vector $z$ that satisfy w.h.p. $\|A - A_{:,j} z^T\|_0 \leq (2 + \epsilon)\mathrm{OPT}^{(1)}$. The algorithm also computes a value $Y$ satisfying w.h.p. $(1 - \epsilon)\mathrm{OPT}^{(1)} \leq Y \leq (2 + 2\epsilon)\mathrm{OPT}^{(1)}$.*

The only steps for which the implementation details are not immediate are Steps 2.2 and 3. We will discuss them in Sections 4.1 and 4.2, respectively. Note that the algorithm from Theorem 8 selects a column $A_{:,j}$ and then finds a good vector $z$ such that the product $A_{:,j} z^T$ approximates $A$. We show that the approximation guarantee $2 + \epsilon$ is essentially tight for algorithms following this pattern.

**Lemma 9.** *There exist a matrix $A \in \mathbb{R}^{n \times n}$ such that $\min_z \|A - A_{:,j} z^T\|_0 \geq 2(1 - 1/n)\mathrm{OPT}^{(1)}$, for every column $A_{:,j}$.*

## 4.1 Implementing Step 2.2

The Step 2.2 of Algorithm 2 uses the following sublinear procedure, given in Algorithm 3.

**Lemma 10.** *Given $A \in \mathbb{R}^{m \times n}$, $u \in \mathbb{R}^m$ and $\epsilon \in (0, 1)$ we can compute in $O(\epsilon^{-2} n \log m)$ time a vector $z \in \mathbb{R}^n$ such that w.h.p. $\|A_{:,i} - z_i u\|_0 \leq (1 + \epsilon) \min_{v_i} \|A_{:,i} - v_i u\|_0$ for every $i \in [n]$.*

---

**Algorithm 3**

---

**Input:** $A \in \mathbb{R}^{m \times n}$, $u \in \mathbb{R}^m$ and $\epsilon \in (0, 1)$.

Let $Z \stackrel{\text{def}}{=} \Theta(\epsilon^{-2} \log m)$, $N \stackrel{\text{def}}{=} \mathrm{supp}(u)$, and $p \stackrel{\text{def}}{=} Z/|N|$.

1. Select each index $i \in N$ with probability $p$ and let $S$ be the resulting set.

2. Compute a vector $z \in \mathbb{R}^n$ such that $z_j = \arg\min_{r \in \mathbb{R}} \|A_{S,j} - r \cdot u_S\|_0$ for all $j \in [n]$.

**Return:** vector $z$.

---

## 4.2 Implementing Step 3

In Step 3 of Algorithm 2 we want to compute a $(1 + \frac{\epsilon}{15})$-approximation $Y_j$ of $\|A - A_{:,j}[z^{(j)}]^T\|_0$ for every $j \in \bigcup_{i \in [|\mathcal{S}|]} C^{(i)}$. We present two solutions, an exact algorithm (see Lemma 11) and a sublinear time sampling-based algorithm (see Lemma 13).

**Lemma 11.** *Suppose $A, B \in \mathbb{R}^{m \times n}$ are represented by column adjacency arrays. Then, we can compute in $O(\|A\|_0 + n)$ time the measure $\|A - B\|_0$.*

For our second, sampling-based implementation of Step 3, we make use of an algorithm by Dagum et al. [11] for estimating the expected value of a random variable. We note that the runtime of their algorithm is a random variable, the magnitude of which is bounded w.h.p. within a certain range.

**Theorem 12.** *[11] Let $X$ be a random variable taking values in $[0, 1]$ with $\mu \stackrel{\text{def}}{=} \mathbb{E}[X] > 0$. Let $0 < \epsilon, \delta < 1$ and $\rho_X = \max\{\mathrm{Var}[X], \epsilon\mu\}$. There is an algorithm with sample access to $X$ that computes an estimator $\tilde{\mu}$ in time $t$ such that for a universal constant $c$ we have:*
*i) $\Pr[(1 - \epsilon)\mu \leq \tilde{\mu} \leq (1 + \epsilon)\mu] \geq 1 - \delta$,   and   ii) $\Pr[t \geq c\,\epsilon^{-2} \log(1/\delta)\rho_X/\mu^2] \leq \delta$.*

We state now our key technical insight, on which we build upon our sublinear algorithm.

**Lemma 13.** *There is an algorithm that, given $A, B \in \mathbb{R}^{m \times n}$ with column adjacency arrays and $\|A - B\|_0 \geq 1$, and given $\epsilon > 0$, computes an estimator $Z$ that satisfies w.h.p. $(1 - \epsilon)\|A - B\|_0 \leq Z \leq (1 + \epsilon)\|A - B\|_0$. The algorithm runs w.h.p. in time $O(n + \epsilon^{-2} \frac{\|A\|_0 + \|B\|_0}{\|A - B\|_0} \log n\})$.*

We present now our main result in this section.

**Theorem 14.** *There is an algorithm that, given $A \in \mathbb{R}^{m \times n}$ with column adjacency arrays and $\mathrm{OPT}^{(1)} \geq 1$, and given $j \in [n]$, $v \in \mathbb{R}^m$ and $\epsilon \in (0, 1)$, outputs an estimator $Y$ that satisfies w.h.p. $(1 - \epsilon)\|A - A_{:,j}v^T\|_0 \leq Y \leq (1 + \epsilon)\|A - A_{:,j}v^T\|_0$. The algorithm runs w.h.p. in time $O(\min\{\|A\|_0, n + \epsilon^{-2}\psi^{-1}\log n\})$, where $\psi = \mathrm{OPT}^{(1)}/\|A\|_0$.*

To implement Step 3 of Algorithm 2, we simply apply Theorem 14 with $A$, $\epsilon$ and $v = z^{(j)}$ to each sampled column $j \in \bigcup_{0 \leq i \leq \log n + 1} C^{(i)}$.

# 5 Algorithms for Boolean $\ell_0$-rank-1

Our goal is to compute an approximate solution of the Boolean $\ell_0$-rank-1 problem, defined by:

$$\mathrm{OPT} = \mathrm{OPT}_A \stackrel{\text{def}}{=} \min_{u \in \{0,1\}^m,\, v \in \{0,1\}^n} \|A - uv^T\|_0, \quad \text{where } A \in \{0, 1\}^{m \times n}. \tag{3}$$

In practice, approximating a matrix $A$ by a rank-1 matrix $uv^T$ makes most sense if $A$ is close to being rank-1. Hence, the above optimization problem is most relevant when $\mathrm{OPT} \ll \|A\|_0$. In this section, we focus on the case $\mathrm{OPT}/\|A\|_0 \leq \phi$ for sufficiently small $\phi > 0$. We prove the following.

**Theorem 15.** *Given $A \in \{0, 1\}^{m \times n}$ with row and column sums, and given $\phi \in (0, 1/80]$ with $\mathrm{OPT}/\|A\|_0 \leq \phi$, we can compute vectors $\tilde{u}, \tilde{v}$ with $\|A - \tilde{u}\tilde{v}^T\|_0 \leq (1 + 5\phi)\mathrm{OPT} + 37\phi^2\|A\|_0$ in time $O(\min\{\|A\|_0 + m + n, \phi^{-1}(m + n)\log(mn)\})$.*

In combination with Theorem 8 we obtain the following.

**Theorem 16.** *Given $A \in \{0, 1\}^{m \times n}$ with column adjacency arrays and with row and column sums, for $\psi = \mathrm{OPT}/\|A\|_0$ we can compute vectors $\tilde{u}, \tilde{v}$ with $\|A - \tilde{u}\tilde{v}^T\|_0 \leq (1 + 500\psi)\mathrm{OPT}$ in time w.h.p. $O(\min\{\|A\|_0 + m + n, \psi^{-1}(m + n)\} \cdot \log^3(mn))$.*

A variant of the algorithm from Theorem 15 can also be used to solve the Boolean $\ell_0$-rank-1 problem exactly. This yields the following theorem, which in particular shows that the problem is in polynomial time when $\mathrm{OPT} \leq O(\sqrt{\|A\|_0}\log(mn))$.

**Theorem 17.** *Given a matrix $A \in \{0, 1\}^{m \times n}$, if $\mathrm{OPT}_A/\|A\|_0 \leq 1/240$ then we can exactly solve the Boolean $\ell_0$-rank-1 problem in time $2^{O(\mathrm{OPT}/\sqrt{\|A\|_0})} \cdot \mathrm{poly}(mn)$.*

# 6 Lower Bounds for Boolean $\ell_0$-rank-1

We give now a lower bound of $\Omega(n/\phi)$ on the number of samples of any $1 + O(\phi)$-approximation algorithm for the Boolean $\ell_0$-rank-1 problem, where $\mathrm{OPT}/\|A\|_0 \leq \phi$ as before.

**Theorem 18.** *Let $C \geq 1$. Given an $n \times n$ Boolean matrix $A$ with column adjacency arrays and with row and column sums, and given $\sqrt{\log(n)/n} \ll \phi \leq 1/100C$ such that $\mathrm{OPT}_A/\|A\|_0 \leq \phi$, computing a $(1 + C\phi)$-approximation of $\mathrm{OPT}_A$ requires to read $\Omega(n/\phi)$ entries of $A$.*

The technical core of our argument is the following lemma.

**Lemma 19.** *Let $\phi \in (0, 1/2)$. Let $X_1, \ldots, X_k$ be Boolean random variables with expectations $p_1, \ldots, p_k$, where $p_i \in \{1/2 - \phi, 1/2 + \phi\}$ for each $i$. Let $\mathcal{A}$ be an algorithm which can adaptively obtain any number of samples of each random variable, and which outputs bits $b_i$ for every $i \in [1 : k]$. Suppose that with probability at least $0.95$ over the joint probability space of $\mathcal{A}$ and the random samples, $\mathcal{A}$ outputs for at least a $0.95$ fraction of all $i$ that $b_i = 1$ if $p_i = 1/2 + \phi$ and $b_i = 0$ otherwise. Then, with probability at least $0.05$, $\mathcal{A}$ makes $\Omega(k/\phi^2)$ samples in total.*

## Footnotes

*This work has been funded by the Cluster of Excellence "Multimodal Computing and Interaction" within the Excellence Initiative of the German Federal Government.

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
