[Reviews · NeurIPS 2017]

Reviewer 1



The paper studies the problem of l_0-Low rank approximation in which the goal is to approximate an input matrix A (m\times n) with a rank-k matrix A’ whose entry-wise l_0 distance is minimum. The paper provides the first biceriteria approximation algorithm for this problem: for any k > 1 in poly(mn), provide a rank O(k\log(mn)) such that ||A-A’||_0 \leq poly(k log(mn)) \times OPT_k where OP_k denotes the closets rank-k matrix to A. Moreover, they provide a (2+\eps)-approx for the case k=1 in sublinear time and further provide an algorithm for the case of binary matrices which seem to be more applicable in practice. However, the approximation ratio depends on the value of optimal solution and works well when OPT is O(||A||_0). The ratio makes the result a bit questionable. The problem theoretically seems natural and standard; however, its application is not well-motivated. The paper also lacks experiments and solid explanation of the application of problem in practice. The results are clean and standard. However, there are some assumptions for which not sufficient evidences have been provided: • Why do they need to have the assumption that sum of rows and column can be accessed in O(m+n)? (lines 138-140). Assuming ||A||_0 and the adj list access you can easily compute them. Apparently, it is required for Theorem 13 and 14 where one can spend ||A||_0 in running time and compute them and still achieve the same running time guarantees. • The conjecture or the inapproximability result for Boolean l_0 rank-1 seems strong without any evidence. Minor comments: • Line 31: rank-k -> rank-$k$ • Line 33: min(mn^2, mn^2) -> probably min(mn^2, m^2n) • Line 131-137: you can mention incident list access model which is a very natural access model in the area of sublinear time algorithms design • Line 168: S^(0) = [n]  S^(0) = [m] • Line 234: The -> we To sum up, the paper has studied and interesting theoretical problem and provides reasonable progress on that. However, its applicability in practice is not well supported.

Reviewer 2



The paper proposes algorithms for finding a rank-k ell_0-minimizing approximation of a given matrix A. It gives approximation algorithms for the real case when p=1 and p > 1 and it also studies the boolean case. For all these cases the authors provide approximation algorithms (in some cases surprisingly sublinear) and in others bi-criteria approximations. Strengths: -- The algorithms are interesting and the proof techniques involved have some interesting twists. -- The paper is well written. Weakenesses: -- The lack of experimental evaluation makes it hard to see the applicability of these algorithms in practical scenarios. -- It would be interesting to reduce the space occupied for the extensive literature review and provide some experimental evaluation.

Reviewer 3



This paper considers the problem of approximating matrices when no clear metric is present on the data and the l_0 norm is used (the number of non-zero elements in the difference between the original data and the approximation). A low-rank solution is proposed. The paper doesn't seem to give a practical case where this is useful. In fact, "low rank" implies some linearity and hence implicitly a metric, which lay not make a lot of sense if we can't interprete the data with a metric to start with. E.g., for k=1 one can easily make the first row and then one element in each other row of A-A' equal to zero (so 2n-1 elements), but then predicting linearly the values of the other elements seems arbitrarily. Also, if the elements in A are continuous variables drawn from a distribution with no linear properties, the probability that for any low rank approximation A' more than these minimal number of elements of A-A' can be made equal to *exactly* zero (the requirement of the l_0 norm) is equal to 0. In that sense, if the data is drawn randomly from a continuous distribution, with very high probability, OPT^{(k)} is something like (n-k)(m-k). If k is "low", then theorems as Theorem 2 talking about multiples of OPT^{(k)} are rather trivial (because O(k^2) OPT^{(k)} > 2.OPT^{(k)} > mn) for "random" data. So maybe the problem is to "discover" that by some external cause the data is very close already to low rank, in the sense that a lot of elements happen to match exactly the linear low rank relation to other elements. In amongst others line 272, the authors too indicate that the work is most interesting if A is already close to low rank. The mathematical elaboration and randomized algorithms are quite interesting from a theory point of view. Moving some details to the supplementary material, and making a consistent story connecting the presented algorithms to clear learning problems, could make this a nice paper. Some details: * Line 48: supplementary -> supplementary material * Line 244: having two times "w.h.p." in the same statement is not really necessary (but could be a compact way to present the result).